# Learning Generative Image Object Manipulations from Language Instructions

## Abstract

The use of adequate feature representations is essential for achieving high performance in high-level human cognitive tasks in computational modeling. Recent developments in deep convolutional and recurrent neural networks architectures enable learning powerful feature representations from both images and natural language text. Besides, other types of networks such as Relational Networks (RN) can learn relations between objects and Generative Adversarial Networks (GAN) have shown to generate realistic images. In this paper, we combine these four techniques to acquire a shared feature representation of the relation between objects in an input image and an object manipulation action description in the form of human language encodings to generate an image that shows the resulting end-effect the action would have on a computer-generated scene. The system is trained and evaluated on a simulated dataset and experimentally used on real-world photos.

## 1 Introduction

For many human-robot interaction scenarios, a long-term vision has been to develop systems where humans can simply instruct robots using natural language commands. This aim encompasses several sub-challenges in robotics and artificial intelligence, including natural language understanding, symbol grounding, task and motion planning, to name only a few. However, a first step for solving these challenging tasks is to study how to best combine the data representations from different domains, which is still a topic of research.

This paper studies whether a perceptual system can via simulation train a computational model to predict the output of actions. The aim is to generate an output image that shows the *effect* of a certain *action* on objects when the instruction to change an object is given in natural language. Figure 1 shows a simplified visualization of the main idea where the input image contains several objects of different shape, size, and color, and the input instruction is to manipulate one of the objects in some way (move, remove, add, replace). The output of the model is a synthetic generated image that shows the effect that the action had on the scene.

To successfully solve the task of depicting the *effect* of a certain *action*, the model must further address a number of different sub-challenges, including; 1) image encoding, 2) language learning, 3) relational learning, and 4) image generation. The key requirement for implementing these human cognitive processes in computational modeling is the data representation for each of the different required domains and how to combine and use their shared representations.

There are several works in the literature that combines some of the aforementioned sub-challenges for solving problems such as image captioning You et al. (2016), image editing Chen et al. (2018), image generation from text descriptions Reed et al. (2016a), visual question answering Santoro et al. (2017); Yang et al. (2016), 3D reconstructed object conditioned on a 2D image Girdhar et al. (2016); Weber et al. (2018), paired robot action and linguistic translation Yamada et al. (2018), and Vision-and-Language Navigation (VLN) Anderson et al. (2018). However, the challenge of how to combine all the four sub-challenges and learn their shared representations still requires more research.

In this work, we propose a system that combines an image encoder, language encoder, relational network, and image decoder and train it in a GAN setting that conditions on both a source image and the action text description to generate a target image of the scene after the action has been performed.

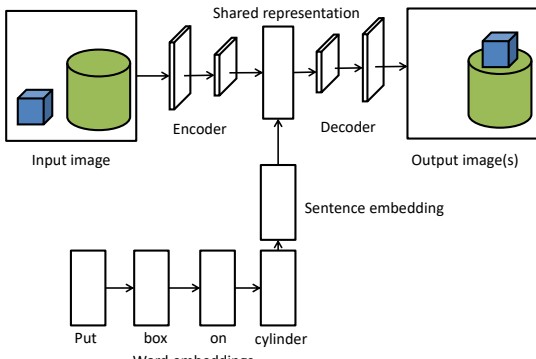

Figure 1: Overview of the proposed method. The input is an initial image and an *action* in the form of language instruction. The output is a generated image that depicts the effect of the performed *action*. The model needs to, first of all, correctly identify the object that should be manipulated and, thereupon, place the object in the correct location by understanding relational concepts.

The system is implemented in PyTorch and trained on a synthetic data-set and evaluated on synthetic and real-world images. The dataset and source code can be downloaded from [link to appear].

## 2 RELATED WORK

The task of generating a target image given a source image and text description has been studied in previous works found in the literature. In work on Language-Based object detection and Segmentation (LBS) Hu et al. (2016b;a), the goal was to identify the regions of the image that correspond to the visual entities described in the natural language description. In the publication on Language-Based Image Editing (LBIE) Chen et al. (2018), the source image was edited according to the text description, where recurrent attentive models are used to fuse image and language features for the subtasks of image segmentation and colorization. The work by Shinagawa et al. (2017) used language instructions (move, expand, or compress) to manipulate 2D images of digits from the MNIST data-set and avatar images to generate the resulting target output image. Our approach builds on the idea of LBIE by extending the input to multiple objects in a 3D space for an object manipulation task (instead of an image editing task). This approach requires, however, also the learning of relations between the different objects.

Generative Adversarial Networks (GAN) Goodfellow et al. (2014) have recently been used extensively to generate realistic images; for example, the works presented by Radford et al. (2015); Dosovitskiy et al. (2015); Yang et al. (2015). In addition, the work by Isola et al. (2017) uses a conditional GAN for pre-defined image-to-image translations. The use of GANs have further been extended to be conditioned on text descriptions to generate images with specific properties Reed et al. (2016a); Nam et al. (2018) or photo manipulation Wang et al. (2018); Bau et al. (2019). As proposed by Zhang et al. (2017), a refinement process that rectifies defects in the first stage, resulting in more realistic synthesized images, can further be achieved by the stacking of GANs.

There are, in addition, works that condition the GAN with additional information (besides the text description) to generate images. The work by Reed et al. (2016b) give control over the location of where the object, or object parts, should be located by providing additional bounding boxes. The work by Dash et al. (2017) also conditioned on the class information to diversify the generated samples. The work by El-Nouby et al. (2018) uses a recurrent GAN to generate iterative output images based on the language instruction.

Moreover, there have been presented works that integrate a recurrent neural network for the image generation. For example, a recurrent variational autoencoder with an alignment model over words and attention mechanisms have been used to generate images from text captions Mansimov et al.

(2016), while a GAN conditioned on an LSTM was, similarly, used for image sequence generation for each word, as presented by Ouyang et al. (2018).

While related works generate output images from text descriptions, they differ from our work in several ways. Firstly, our method conditions on both the input image and the text description for generating the target image. Secondly, we integrate a relational network in the shared representation for learning relations between objects in a 3D simulated space. Thirdly, we perform image object manipulation instead of image editing, and there is only one correct solution for each pair of the source image and text description, which is typically not the case for other works using GANs.

## 3 DATA AND METHOD

### 3.1 DATA GENERATION

The data consists of a generated synthetic data-set consisting of objects that are randomly placed on a table and contains the *before state* image, the *after state* image, and an *action* sentence, as exemplified in Figure 2.

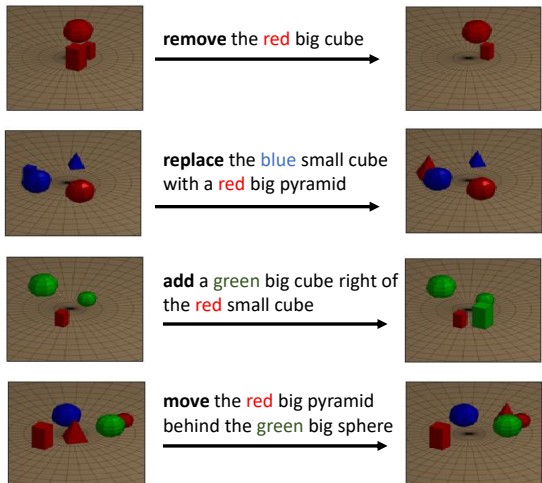

Figure 2: Examples of input (left) and output (right) image for each of the four actions *remove*, *replace*, *add*, and *move*.

The actions are chosen from a set of 4 possible actions: 2 non-relational $\{remove, replace\}$, and 2 relational $\{move, add\}$. A summary of the structure of the actions can be seen in Table 1. The non-relational actions choose one object and either *removes* it or *replaces* it with another object with a specified shape, color, and size. These instructions are non-relational since they do not require to learn the relation between the manipulated object and the referential object in the scene.

Table 1: The four action commands that are explored in this work.

| Action | Type |
|---|---|
| REMOVE {object1} | non-relational |
| REPLACE {object1} WITH {object2} | non-relational |
| MOVE {object1} {on top, left, right, behind, in front} OF {object2} | relational |
| ADD {object1} {on top, left, right, behind, in front} OF {object2} | relational |

The synthesized images are of size $128 \times 128 \times 3$, and each image contains 2 to 6 objects which (each) is either a *box*, *sphere*, or *pyramid*, with the color *red*, *green*, or *blue*, and with the size *small*

or *big*. There are no duplicate objects in the same scene that have the same shape, color, and size in order to avoid ambiguities. For the training data, 1000 input images were generated with 10 *action* sentences for each image. The validation and test data each consisted of 200 images with 10 actions per image.

## 3.2 OVERVIEW OF METHOD

The proposed method, called CAE+LSTM+RN, can be seen in Figure 3 and consists of an image encoder, language encoder, relational network, and an image decoder.

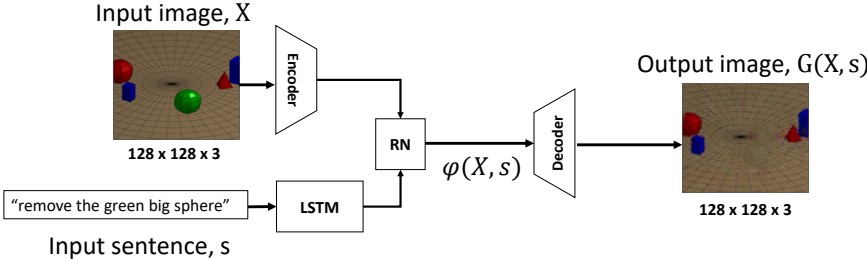

Figure 3: Detailed overview of the proposed method. A CNN-based encoder is used to represent the input image $X$ and the last output from a LSTM is used as representation of the input sentence $s$. A relational network combines these two representations into a shared representation, $\varphi(X, s)$. The generated output image is obtained with an image decoder.

### 3.2.1 IMAGE ENCODER

The image encoder follows a similar structure as DCGAN Radford et al. (2015), with 4 convolutional layers with stride 2, padding 1, and filter size 4 in each layer and numbers of filters 64, 128, 256, and 512 in each layer. Each convolutional layer is followed by batch normalization and relu-activation. The input images are of size $128 \times 128 \times 3$, and the output of the image encoder is of size $8 \times 8 \times 512$.

### 3.2.2 LANGUAGE ENCODER

The last output of a 2-layered bi-directional LSTM, with 24 hidden units, was used for the sentence embedding, $g(s)$. A word embedding with dimension 30 was learned from a vocabulary of size 17, consisting of the following words: "move, remove, replace, add, small, big, pyramid, sphere, cube, red, green, blue, front, behind, left, right, and top". Each sentence was also initially parsed and pre-processed such that articles and prepositions (e.g., words such as "in, on, of, the"), could be removed in an early stage. The sentences were presented in reverse word order.

### 3.2.3 RELATIONAL NETWORK

A Relational Network (RN) Santoro et al. (2017) is used to both learn relations between object-pairs, as well as merging image and language representations. The output of the image encoder, $f(X)$, is of size $8 \times 8 \times 512$, where each filter response at position $i$ and $j$ in $f(X)_{(i,j,:)}$ is considered one object, resulting in a total of $8 * 8 = 64$ objects. An object-pair vector is made by concatenating two object vectors, with their spatial coordinates, and with the sentence representation $[f(X)_{(i,j,:)}^T ij f(X)_{(k,l,:)}^T kl g(s)]$, resulting in $64 \times 64 = 4096$ object-pairs. Each object-pair vector is feed through a three-layered fully-connected network with 256 hidden units and relu-activation. The output of the last layer for each object-pair is summed up and reshaped to size $8 \times 8 \times n_{RN}$ where $n_{RN}$ is set to 32.

### 3.2.4 IMAGE DECODER

The image decoder first performs a convolution with 512 filters and stride 1, padding 1, and filter size 3 then follows a similar structure as the image encoder with the same stride, padding, and filter size but with reversed number of filters in each transposed convolutional layer. The output of the image decoder is $128 \times 128 \times 3$.

## 3.3 INTEGRATION OF GENERATIVE ADVERSARIAL NETWORK (GAN)

The integration of the proposed model with a GAN, called CAE+LSTM+RN+GAN, can be seen in Figure 4. The generator has the same structure as the proposed model from Section 3.2. Since there is only one correct solution for each image-sentence pair, the generator does not use any random number as input. The loss function for the generator is:

$$L_G = \log(1 - D(\hat{Y}, \varphi(X, s)) + \alpha * ||h(\hat{Y}) - h(Y)||_2^2 + \beta * ||\hat{Y} - Y|| + \gamma * ||X - Y||_2^2 \quad (1)$$

where the first term is the generator loss, the second is the feature matching loss, the third is L1-loss between fake and real image and the fourth is the L2-loss between the input image and the output real image. The last term is to focus on the pixels that have been changed in order to avoid getting stuck in a pure reconstruction. The trade-off parameters $\alpha$, $\beta$, and $gamma$ are set to 100, 100, and 1000 respectively.

The discriminator first uses an image encoder on either the generated image $\hat{Y}$, the ground truth image $Y$, or a randomly sampled wrong output image $\tilde{Y}$. The encoder uses the same structure as the image encoder in the generator except with leaky relu-activation instead of relu. The output from the encoder is concatenated with the shared representation $\varphi(X, s)$ from the generator. A classifier consisting of a convolutional layer with stride 2, padding 1, filter size 4 and 256 filters, batch norm, leaky relu followed by a convolutional layer with stride 1, padding 0, filter size 4 and 1 filter and sigmoid is used to classify if the output image is real or fake/wrong. The loss function for the discriminator is:

$$L_D = \log(D(Y, \varphi(X, s)) + \log(1 - D(\hat{Y}, \varphi(X, s)) + \log(1 - D(\tilde{Y}, \varphi(X, s)) \quad (2)$$

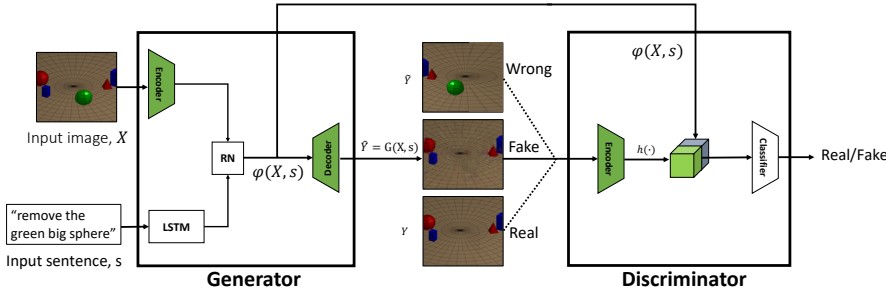

Figure 4: Integration of the model with a GAN. The discriminator is conditioned on the shared representation of the input image and sentence.

## 4 EXPERIMENTAL RESULTS

### 4.1 EXPERIMENTAL SETUP

The two models CAE+LSTM+RN and CAE+LSTM+RN+GAN are trained on the training set of 10000 image input-output pairs using the Adam optimizer with a learning rate of $2e^{-4}$, $\beta_1$ 0.5, $\beta_2$ 0.999, and minibatch size of 25 on a machine with GeForce GTX 1070 8 GB, Intel i7-8700K@3.7GHz, and 32 GB RAM. The model was evaluated visually and by calculating the root-mean-square-error (RMSE) on a test set of 200 images.

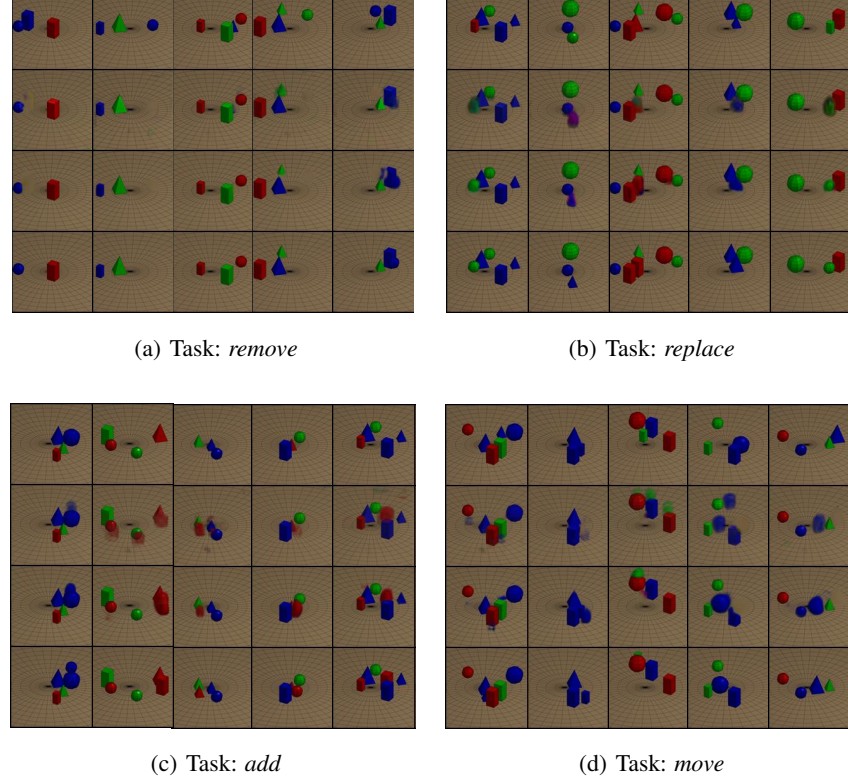

(a) Task: *remove*                    (b) Task: *replace*

(c) Task: *add*                    (d) Task: *move*

Figure 5: Results from two models. Five input images (columns) for each of the four actions (a-d). The first row in each sub-figure is the input image data and the fourth row is the target output image. The second and third row is the generated output from the CAE+LSTM+RN and CAE+LSTM+RN+GAN model, respectively. The action sentence can be inferred by comparing the difference between the first and fourth column.

## 4.2 IMAGE OUTPUT GENERATION

The generated output from the two models can be seen in Figure 5. Here, the *rows* represent the input image, generated output from CAE+LSTM+RN, generated output from CAE+LSTM+RN+GAN, and the target output image, respectively.

The GAN model is able to reduce the noise for all four tasks and the copies of objects sometimes generated with the add and remove task. Some colors that are not present in the training set where corrected with GAN, e.g., a purple object in the task replace. However, the shape of the object are sometimes not clear, especially for cubes that looked liked large spheres and all small objects look like small cube.

A summary of the root-mean-square-error (RMSE) between the generated and target images on the 200 test images can be seen in Table 2. The first row is a baseline model that only uses the image data but no action sentence information. The tasks with the lowest RMSE are the non-relational tasks *remove* and *replace*.

Table 2: Comparison of the RMSE between the generated image and the target image on the test data between different models.

| Model | *remove* | *replace* | *add* | *move* | Overall |
|---|---|---|---|---|---|
| CAE | 0.0407 | 0.0482 | 0.0441 | 0.0519 | 0.0457 |
| CAE+LSTM+RN | 0.0144 | 0.0222 | 0.0281 | 0.0264 | 0.0229 |
| CAE+LSTM+RN+GAN | 0.0134 | 0.0208 | 0.0272 | 0.0249 | 0.0221 |

### 4.3 EXPERIMENTS ON REAL-WORLD DATA

The pre-trained CAE-LSTM-RN-GAN model on simulated data was used on a sequence of five real-world images with four action sentences and six objects, see Figure 7.

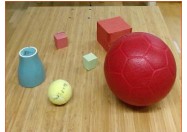 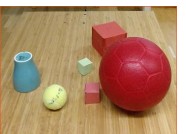 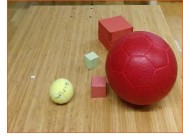 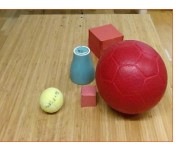 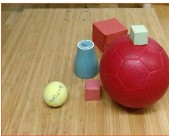

Figure 6: Real-world images with four action sentences. The action sentences where: "move the red small cube left of red big sphere", "remove the blue big pyramid", "replace the green small cube with a blue big pyramid", and "add a green small cube on top of red big sphere".

For pre-processing, a manual segmentation on each object and the background is performed on the first image to get an average color. A color-based segmentation based on the average color for each object is performed on all five real-world images. Each image is converted to a HSV representation. The hue values for pixels segmented as object 1,3,6 are changed to 0 (red), object 2 and 4 to $1/3$ (green), object 5 to $2/3$ (blue), and the background as 0.09, which is the average hue value for the training images. The saturation is set to 1 for the objects and 0.3894 for the background, which is also taken from training images. The value (in HSV) for each pixel is multiplied by a factor so that the average value for each real-world image is the same as the average of the training images, which was 0.4389. The new HSV representation is then converted back to RGB representation. This minimal pre-processing step made the real-world images look more like the training images.

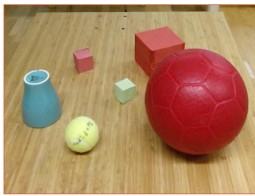 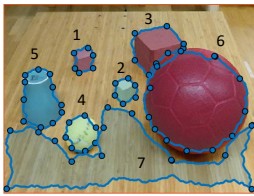 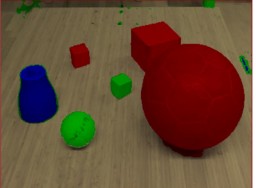

Figure 7: Pre-processing of real-world images. (left) Original real-world image with 6 objects. (middle) manual segmentation of the 6 objects and the background. (right) Adjustment of HSV-values based on automatic color-based segmentation.

The results can be seen in Figure 8. For the first action (column 1) to "move the red small cube left of red big sphere", the model attempts to remove the red small cube and generates an object with the right color and size, but with the wrong shape and arguable the location. An explanation to the faulty location might be that the sphere is much larger than the large sphere in the training data. The second action (column 2) is to "remove the blue big pyramid", which in reality is an upside-down coffee cup. The model partly removes the correct object. The third action (column 3) to "replace the green small cube with a blue big pyramid" replaces the correct object with an object of correct color but with the wrong shape. The last action (column 4) is to "add a green small cube on top of red big sphere".

## 5 CONCLUSIONS

This work has combined an image encoder, language encoder, relational network, and image generator to visualize the effect an action would have on a simulated scene. The model was used in a GAN learning setting to improve the generated images. The focus in this work has been on learning meaningful shared image and text representations for relational learning and object manipulation.

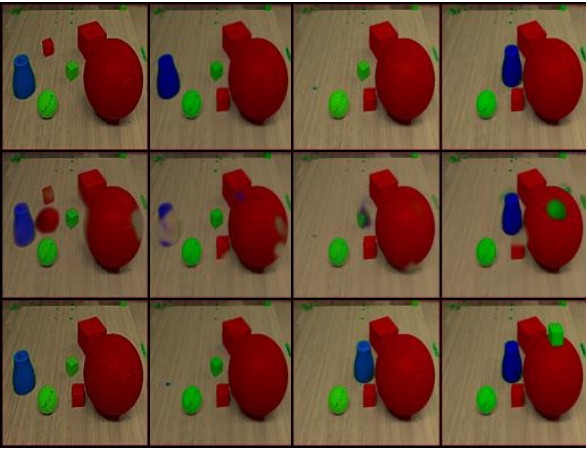

Figure 8: Results on real-world images with pretrained CAE-LSTM-RN-GAN model on simulated images. First row show the pre-processed input image, second row show the generate output image, and third row show the target image.

Directions for future work include generating sequenced images, training on real-world images, adapt the system to arbitrarily text instructions, recognising objects from a pre-trained image encoder, using different image sizes, and implementing the system on a real robot. An initial step in any of these directions for future work could alternatively be to train a shared image and text representations on a similar synthetic data-set, but with more realistic objects (e.g., the CLEVR dataset Johnson et al. (2017)), and, subsequently, transfer and fine-tune the model to a real-world setting.

ACKNOWLEDGMENTS

Hidden during review

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
