# OpenReview forum: "Learning Generative Image Object Manipulations from Language Instructions"
_ICLR.cc/2020/Conference — Reject_

### Official Review · AnonReviewer1 · 2019-10-24
**Official Blind Review #1**

**Rating:** 3

**Review:**

This paper proposes a model that takes an image and a sentence as input, where the sentence is an instruction to manipulate objects in the scene, and outputs another image which shows the scene after manipulation. The model is an integration of CNN, RNN, Relation Nets, and GAN. The results are mostly on synthetic data, though the authors also included some results on real images toward the end.

This paper, despite studying an interesting problem, is limited in terms of its technical innovations and experimental results. My recommendation is a clear reject.

The model is simply an integration of multiple standard neural nets. To me, it's unclear how the system can inspire future research. Such an integration of neural nets won't generalize well. The authors have to pre-process real images in a very specific way for limited sim-to-real transfer. It's unclear how the model can work on more complex images, nor to mention scenes or sentences (or actions) beyond those available during training.

The experimental setup is very simple. The model is tested on scenes with a clean background and a few geometric primitives. There are only four actions involved. There are no comparisons with published, SOTA methods. All experiments are with the ablated model itself. Considering all this, I believe this paper cannot be accepted to a top conference such as ICLR.



**Experience Assessment:**

I have published in this field for several years.

**Review Assessment: Checking Correctness Of Derivations And Theory:**

I assessed the sensibility of the derivations and theory.

**Review Assessment: Checking Correctness Of Experiments:**

I assessed the sensibility of the experiments.

**Review Assessment: Thoroughness In Paper Reading:**

I read the paper thoroughly.

---

### Official Review · AnonReviewer2 · 2019-10-24
**Official Blind Review #2**

**Rating:** 3

**Review:**

This paper proposes an architecture for generating images with objects manipulated according to user-specified or conditional instructions. I find the domain very interesting and do believe that tasks like these are critical for learning human-like cognitive capabilities.

This paper is also very clear and easy to follow and understand what the authors have done.

But I do feel like this work could use more polishing. There are four components that are used in this work, CVAE, LSTM, RN, and GANs. It seems that those components are all taken straight out of the shelf and combined. It would be interesting to see what subtle changes were important in a combined system to further increase performance.
For example, why is RN only before decoding, could RN possibly help the decoder as well?

What are some of the most frequent failure cases?
The qualitative results look reasonable, and I’m quite surprised that only 10K images were used for training. Improvements in which areas would lead to perfect results?

Would better performance be obtained if every module were to be trained separately first, rather than the proposed end-to-end approach?


**Experience Assessment:**

I have read many papers in this area.

**Review Assessment: Checking Correctness Of Derivations And Theory:**

N/A

**Review Assessment: Checking Correctness Of Experiments:**

I assessed the sensibility of the experiments.

**Review Assessment: Thoroughness In Paper Reading:**

N/A

---

### Official Review · AnonReviewer3 · 2019-10-26
**Official Blind Review #3**

**Rating:** 1

**Review:**

1. The paper aims to train a model to move objects in an image using language. For instance, an image with a red cube and blue ball needs to be turned into an image  with a red cube and red ball if asked to "replace the red cube with a blue ball". The task itself is interesting as it aims to modify system behavior through language.

The approach the authors  take is to encode an image with a CNN, encode the sentence with an RNN and use both representations to reconstruct  the frame (via a relational network and decoder) that solves this task. This process as described was already done in (Santoro 2017). The idea of using a CNN feature map and LSTM embedding to solve spatial reasoning tasks is not new.

The main contribution is to add a discriminative loss to turn the problem into a "is this solution correct or not." This is interesting but does not perform much better than the baseline of not using the GAN loss (as suggested by the results in Table 2). This suggests that the GAN term is not adding as much value as the authors claim.

2. Reject
- Reason 1: The results in Table 2 show that the GAN does slightly better (0.0134 vs 0.0144) in RMSE against the non-GAN version. This improvement does not seem statistically significant enough to warrant the added GAN complexity.
- Reason 2: Other baselines need to be considered, AE, VAE or other variations.
- Reason 3: No ablations on the impact of the parameters to eq 1.

3. To improve the paper I suggest adding other baselines such as VAE, AE. In addition, consider using more negative samples instead of the single negative image.

**Experience Assessment:**

I have read many papers in this area.

**Review Assessment: Checking Correctness Of Derivations And Theory:**

N/A

**Review Assessment: Checking Correctness Of Experiments:**

I assessed the sensibility of the experiments.

**Review Assessment: Thoroughness In Paper Reading:**

I read the paper at least twice and used my best judgement in assessing the paper.

---

### Decision · Program_Chairs · 2019-12-19

**Decision:**

Reject

**Comment:**

The submission proposes to train a model to modify objects in an image using language (the modified image is the effect of an action). The model combines CNN, RNN, Relation Nets and GAN and is trained and evaluated on synthetic data, with some examples of results on real images.

The paper received relatively low scores (1 reject and 2 weak rejects).  The authors did not provide any responses to the reviews and did not revise their submission.  Thus there was no reviewer discussion and the scores remained unchanged.

The reviewers all agreed that the submission addressed an interesting task, but there was no special insight in how the components were put together, and the work was limited in the experimental results.  Comparisons against additional baselines (AE, VAE), and ablation studies or examinations of how the components can be varied is needed.

The paper is currently too weak to be accepted at ICLR.  The authors are encouraged to improve their evaluation and resubmit to an appropriate venue.